# Nitrogen fertilizer application rate impacts eating and cooking quality of rice after storage

**Hanling Liang**[1], **Dongbing Tao**[1], **Qi Zhang**[1], **Shuang Zhang**[1], **Jiayi Wang**[2], **Lifei Liu**[1], **Zhaoxia Wu**[1]*, **Wentao Sun**[3]*

**1** College of Food Science, Shenyang Agricultural University, Shenyang, People's Republic of China,
**2** College of Food and Chemical Engineering, Shaoyang University, Shaoyang, People's Republic of China,
**3** Institute of Plant Nutrition and Environmental Resources, Liaoning Academay of Agricultural Sciences, Shenyang, People's Republic of China

* wuzxsau@163.com (ZW); wentaosw@163.com (WS)

**Data Availability Statement:** All relevant data are within the manuscript and its Supporting Information files.

## Abstract

The effect of nitrogen fertilizer application on the quality of rice post-storage is not well understood. The eating and cooking quality (ECQ) of rice treated with 0 (CK, control), 160 (IN, insufficient nitrogen), 260 (AN, adequate nitrogen), and 420 (EN, excessive nitrogen) kg N/ha was analyzed over 12 months of storage. Results showed that the rate of nitrogen fertilizer application had no significant impact on the changes in taste value during storage. However, EN application significantly increased the hardness ($p < 0.05$), reduced the gumminess ($p < 0.05$), and delayed the decline in the viscosity of rice paste by two months after one-year storage, compared with other treatments. In conclusion, although EN application resulted in an inferior texture of rice, it delayed the quality change by two months during storage. It was demonstrated that a rational nitrogen application rate (0–260 kg N/ha) for rice cultivation is particularly important to obtain high ECQ; however, EN may be beneficial for the stability of the ECQ during storage.

## Introduction

Rice, as a staple food, plays an important role in the human diet and is usually stored for a prolonged period after harvest to meet the needs of people owing to its seasonal growth. However, several studies have reported that the storage process may result in the deterioration of the eating quality of the rice, such as increased hardness and reduced viscosity [1]. Furthermore, the cooking and gelatinization characteristics may be changed, including an increase in water absorption, volume expansion rate, cooking time, and setback, as well as a decrease in trough viscosity, final viscosity, and breakdown [2]. This deterioration is mostly considered to be related to changes in the interactions between the proteins and starch within the rice grain [3–7].

Countless efforts have been made in recent years to reduce the deterioration of rice quality during the storage period. Low temperature and humidity [8,9], as well as vacuum or nano packaging [10], have been proven to be beneficial in maintaining rice quality during storage. Such methods have been applied in milled rice that has a higher commodity price [11,12]; however, such methods have not been widely applied in the storage of paddy rice, owing to the

**Funding:** This work was supported by grants from the National Key Research and Development Program of China (Grant No. 2018YFD0300300), and Science and Technology Innovation Talents Training Project of Liaoning Province (Grant No. XLYC1802044).

**Competing interests:** The authors have declared that no competing interests exist.

trade-off between the high cost of those storage methods and the relatively low commodity value of paddy rice. Thus, paddy rice is usually stored under natural conditions [13]. Thus far, no feasible solution is available to counter the problem of quality loss during paddy rice storage. In addition to storage conditions, the initial quality of rice is also a key factor for the end-use quality of rice after storage. Fertilization, especially the nitrogen fertilizer application rate, is important for the formation of the initial quality of rice. Nitrogen fertilizers have a prominent effect on the eating and cooking quality (ECQ) of rice [14], mainly because their application increases the protein content and decreases the amylose content of rice [15], thereby promoting an increase in the hardness of rice, decreasing its palatability [16–18], and altering the rice gelatinization characteristics and cooking quality.

The effect of nitrogen application rate on the initial quality of fresh rice is well known, but whether this effect persists during a period of storage remains unclear. Previous studies have indicated that fertilization exerts certain effects on the quality of tomato and potato during storage [19,20]. However, different nitrogen fertilization levels affecting rice quality during storage remain understudied. Here, we explored the dynamic changes in the ECQ during the storage of rice produced using different nitrogen application rates and discussed the likely reasons underlying the variations in ECQ after storage, to provide a theoretical basis for the production and quality control of high-quality rice from the field to the table.

## Materials and methods

### Materials

The experiment was conducted at Liaohe Delta, Panjin, China (122°14′17″N, 41°9′31″E). The physical and chemical properties of the 0–20 cm soil were as follows: pH 8.2, organic matter 2.26%, total nitrogen 0.14%, alkali nitrogen 10.52%, available phosphorus 0.002%, available potassium 0.016%, and bulk density 1.39 g/cm$^3$. The following treatment groups were established based on our previous research, in which the effect of nitrogen application rates (0, 160, 210, 260, 315, and 420 kg N/ha) on rice quality was investigated, and due to significant effects on the ECQ of rice: a control group without nitrogen treatment (CK; 0 kg N/ha) and three treatment groups with insufficient (IN; 160 kg N/ha), adequate (AN; 260 kg N/ha), and excessive nitrogen (EN; 420 kg N/ha) treatment. The typical nitrogen fertilization of 260 kg N/ha was usually used by local farmers. Except for the nitrogen application rates, standard practices for rice cultivation were followed by local farmers. The size of each subplot was 50 m$^2$. The transplanting density was 30 cm × 18.2 cm with three seedlings placed in each hill. The experiment was conducted in three replicates. The rice cultivar used was Yanfeng 47, which is one of the main local varieties. The seedlings were transplanted on May 25 and were harvested on October 8 in 2018.

### Sample preparation

All harvested grains were air-dried for a month to reduce moisture content to approximately 14%; each paddy rice (500 g) was packed in a nylon net bag, placed in a carton, and stored under laboratory conditions for 12 months. The temperature and humidity conditions in the laboratory were recorded every 3 days (Fig 1). The paddy rice was processed at an interval of 2 months in a ridge mill (FC2K, Yamamoto, Japan) and a milling machine (VP-32T, Yamamoto, Japan) to obtain brown and milled rice, respectively, for subsequent analyses.

### Chemical component analysis

The protein and amylose contents of brown rice were measured using a near-infrared grating nutrient analyzer (DA7200, Perten, Sweden). Fat content was determined by adopting the

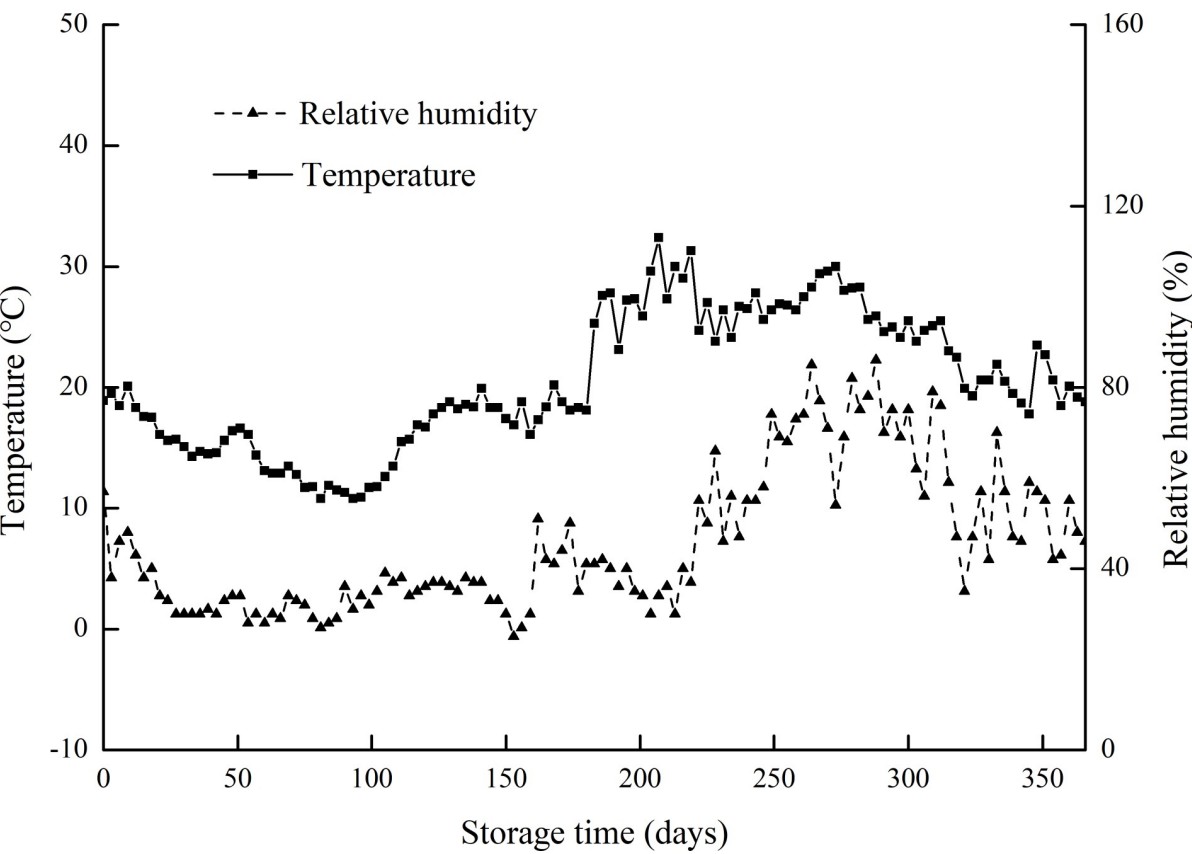

**Fig 1. Record of temperature and relative humidity in the laboratory during the experimental period from November 2018 to November 2019.**

Soxhlet extraction method described in GB 5009.6–2016 using 1.000 g brown rice flour. Rice moisture was determined using the constant weight method at 105˚C in an oven as per methods described in GB 5009.3–2016.

### Eating quality analysis

The taste value of the cooked rice was assessed according to methods that were modified based on those of Champagne et al. (1996) [21]. Briefly, 30 g of milled rice was cooked according to GB/T 15682–2008 guidelines; then, 8 g of the cooked rice was pressed into rice cakes, and the taste value was measured using a rice taste analyzer (STA1B, Satake, Japan). The texture of the rice cakes (hardness, gumminess, and springiness, defined for instrumental texture analyses as per the study published by Champagne et al. (1999) [22]), was determined using the Texture Analyzer (Brookfield Engineering Laboratories, MA, US) following the methods described by Zhang et al. (2019) [23]. The sample was compressed using a 35 mm global probe attachment at a speed of 2 mm/s. The texture profile analysis settings were as follows: pre-test speed, 2.00 mm/s; post-test speed, 2.00 mm/s; time, 10 s; trigger force, 0.05 N. The measurement was conducted in 10 replicates.

### Cooking quality analysis

The cooking quality was determined by following the method prescribed by Gujral and Kumar (2003) [24], with minor modifications. Briefly, 2.5 g of milled rice was boiled at 100˚C

for 10 min in a beaker containing 50 mL distilled water. Then, three grains of rice were removed every minute and pressed between two glass slides until the grains showed no chalky core; the total time was recorded as the cooking time, and the measurement was done in three replicates. For determining volume expansion and water uptake, we measured the volume and weighed 2.5 g of milled rice before and after boiling. The pH of rice soup was determined using a pH meter (PB-10, Sartorius, Germany). Volume expansion and water uptake were calculated using the following equations:

$$\text{Volume expansion (\%)} = \frac{\text{Volume of cooked rice} - \text{Volume of uncooked rice}}{\text{Volume of uncooked rice}} \times 100$$

$$\text{Water uptake (\%)} = \frac{\text{Weight of cooked rice} - \text{Weight of uncooked rice}}{\text{Weight of uncooked rice}} \times 100$$

## Pasting characteristic analysis

The pasting parameters of milled rice flour, such as peak viscosity, trough viscosity, breakdown, final viscosity, setback, peak time, and pasting temperature (Fig 2), were determined using a rapid viscosity analyzer (RVA-4, Newport Scientific, Australia). Milled rice flour (3 g) was passed through a 100-mesh screen into a sample box containing 25 mL distilled water and was then analyzed using the rapid viscosity analyzer.

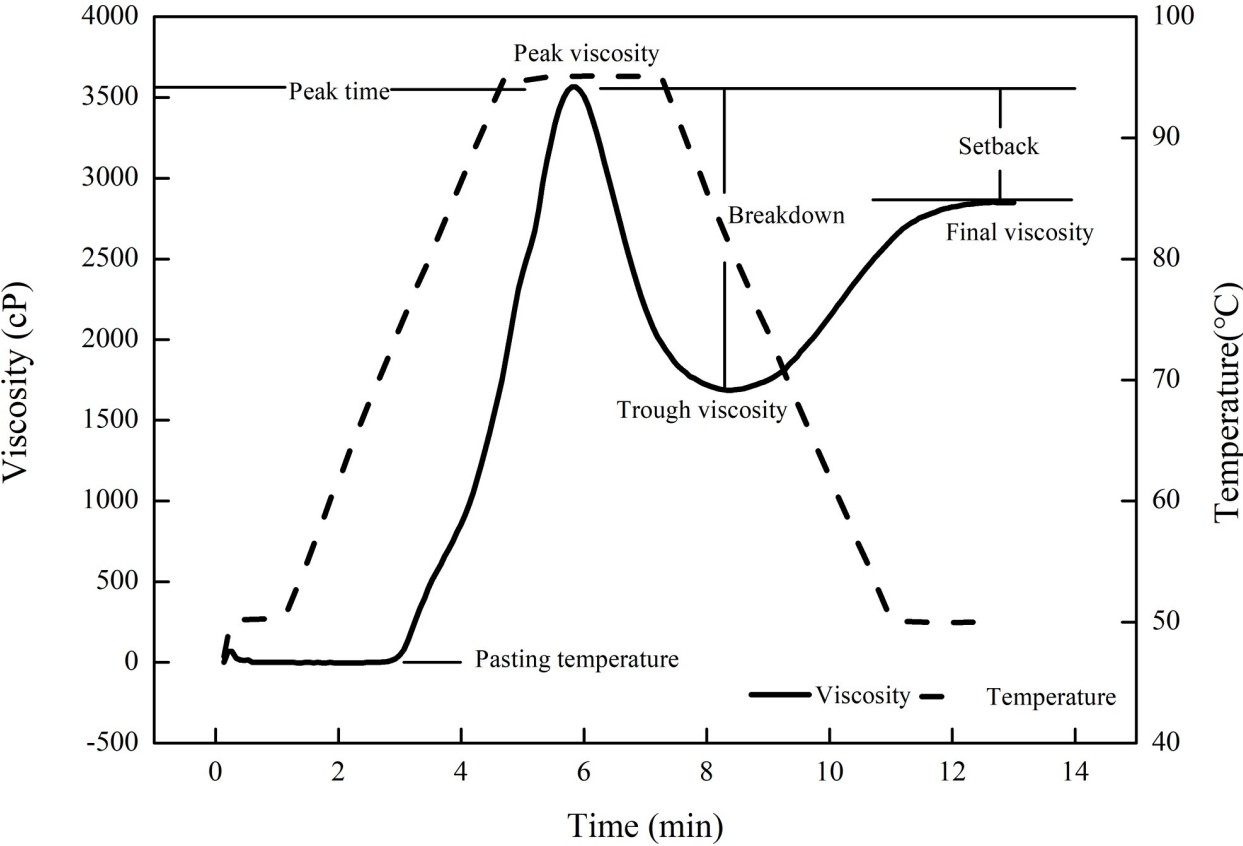

**Fig 2. Rapid viscosity analyzer profile of milled rice flour.**

## Statistical analysis

All data were processed using Microsoft Excel 2019 and then analyzed for significance using the SPSS 22 software with Duncan's test ($p < 0.05$). All tables were generated using Microsoft Word 2019. Graphs were plotted using Origin 8.5. Analysis of variance and linear correlation analysis were conducted using SPSS 22, and linear regression was used for Pearson's correlation analysis between rice quality and chemical components during storage.

## Results and discussion

### ECQ and chemical composition of rice

Among the ECQ indices evaluated, nitrogen application rates and storage time showed significant interaction effects on gumminess, trough viscosity, and setback ($p < 0.05$; S1 Table), indicating that nitrogen application rate exerted a significant impact on the ECQ of rice during postharvest storage. Thus, controlling the nitrogen application rate in the fields is important for maintaining the ECQ of rice during storage.

### Content of chemical components

Nitrogen application did not impact the change trend of protein, fat, and moisture contents in rice during storage (Fig 3). The change in rice content before and after storage under the four nitrogen application treatments had a range of 8%–12% for protein, 5%–11% for fat, and 29%–30% for moisture. Amylose content in the CK and IN samples did not vary significantly during storage; however, EN treatment significantly exacerbated the rate of amylose decline during the period of storage, spanning from 4 to 10 months. During the storage period of 4–10 months, the amylose content decreased by 1% in the CK samples and 4% with EN treatment, which is approximately four times that of CK. This finding demonstrated that nitrogen application in the field changed the starch metabolism of rice during storage.

The decline in amylose content of rice during storage was also observed by Gu et al. [25] and was attributed to endogenous degradation. α-Amylase is the main enzyme involved in starch hydrolysis. It has been reported that the activity of α-amylase in rice is higher under higher nitrogen application rates [26]. Therefore, for our study, it is possible that nitrogen application increased the activity of α-amylase in rice and accelerated the degradation of amylose during storage. However, this effect seems to be dependent on nitrogen application rate because only EN was significant ($p < 0.05$). Moreover, because amylose content is closely related to the ECQ of rice [27,28], the variations in amylose content between the different nitrogen application rates may have contributed to the differential changes in rice ECQ during storage.

### Eating quality

**Taste value.**   Rice exposed to different nitrogen application rates showed similar variation trends in taste value during storage (Fig 4). The taste value of rice was significantly reduced with a higher nitrogen application rate and slightly decreased after storage, which is consistent with previous studies [17,29]. Moreover, without storage, the taste value of rice under EN treatment significantly decreased from 63 to 48 compared with the CK group. For the CK group, the taste value of rice stored for 12 months slightly decreased from 63 to 60 compared with that observed without storage (Fig 4). Notably, nitrogen application had a greater impact on the taste value of rice than storage under the conditions of this study. Accordingly, controlling the rate of nitrogen application for cultivating rice is particularly important for maintaining the high eating quality of rice during storage.

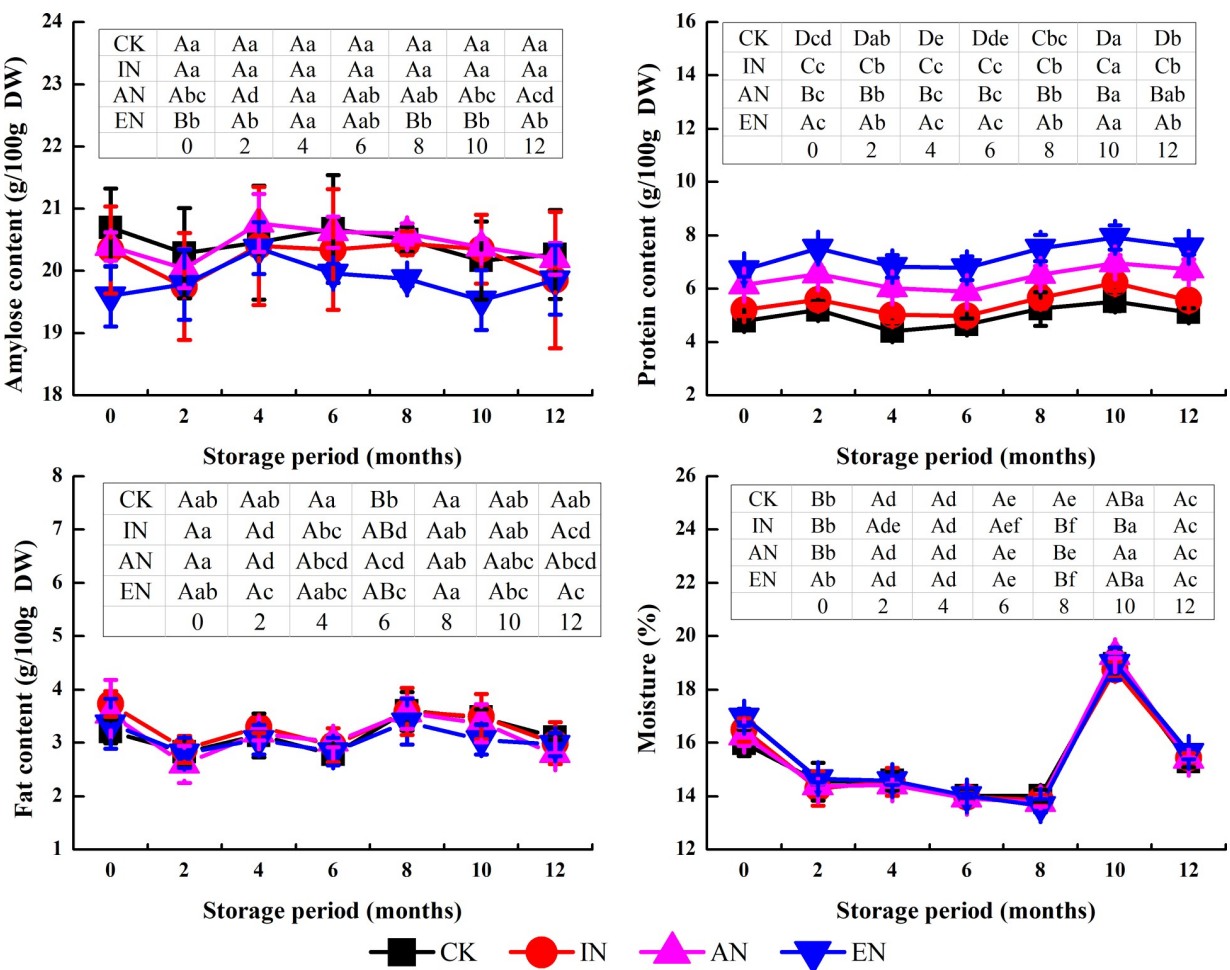

**Fig 3. The changes in the content of chemical components in rice under storage, following the different nitrogen application rates.** CK, control group (0 kg N/ha); IN, insufficient nitrogen (160 kg N/ha); AN, adequate nitrogen (260 kg N/ha); EN, excessive nitrogen (420 kg N/ha). Data (mean ± standard deviation, n = 9) with different letters are significantly different (p < 0.05). For each parameter in the inset table, different lowercase letters in the same rows differ significantly as a function of storage time (Duncan's test; p < 0.05). Different uppercase letters in the column denote significant differences as a function of nitrogen application rates (p < 0.05).

**Textural characteristics.**　Hardness, gumminess, and springiness are important parameters for the evaluation of textural characteristics of cooked rice [30]. In this study, nitrogen application increased the hardness and reduced the gumminess but had little impact on the springiness of cooked rice, as compared before and after storage (Fig 4), which indicated that nitrogen application accelerated the degree of deterioration of textural characteristics of rice during storage.

The changes in hardness of rice after storage for 12 months for the CK, IN, AN, and EN groups was -11%, +16%, +11%, and +14%, respectively, compared with rice without storage (Fig 4). Further, the change in gumminess of rice from the CK, IN, AN, and EN groups was -40%, +16%, -47%, and -48%, respectively.

Hardness is positively related to the amylose content [27], and gumminess is markedly related to the amylopectin content in the leachate produced by cooking the rice [31], especially the high proportion of short-chain amylopectin [27]. However, in our study, the more profound increase in hardness and decline in gumminess for 0–12 months of storage under EN treatments seemed to have little connection to the changes in the chemical components alone

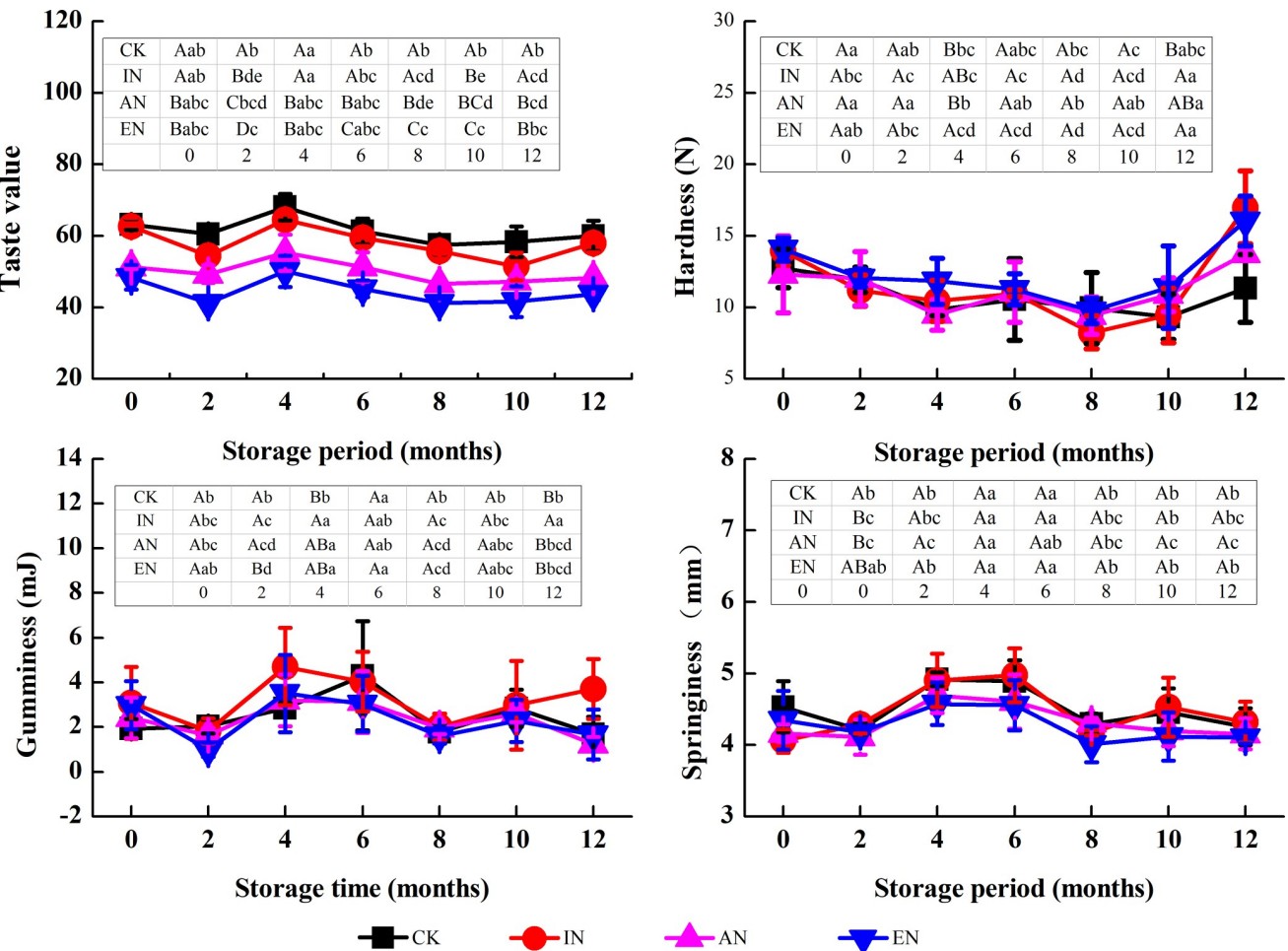

**Fig 4. The changes in the eating quality of rice during storage, following the different nitrogen application rates.** CK, control group (0 kg N/ha); IN, insufficient nitrogen (160 kg N/ha); AN, adequate nitrogen (260 kg N/ha); EN, excessive nitrogen (420 kg N/ha). For each parameter, the mean ± standard deviation (n = 3) followed by different lowercase letters in the same rows differ significantly as a function of storage time (Duncan's test; p < 0.05). Different uppercase letters in the column denote significant differences as a function of different nitrogen application rates (p < 0.05).

(protein, amylose, fat, and moisture contents), as shown in Fig 3. Moreover, the increase in hardness and decline in gumminess during storage is mainly due to the formation of amylose–lipid complexes [32] and the increase in the content of high molecular weight proteins caused by protein aggregation [33]. Further, the solubility of protein and amylose is reduced because of changes in their interactions in rice and results in reduced gumminess [1]. However, the decreased amylose content observed in our study may not be conducive to the formation of amylose–lipid complexes. Kaur (2016) [14] showed that nitrogen application increases the high molecular weight subunits in protein fractions. Thus, in this experiment, it is possible that the nitrogen application had a greater effect on the increase in the content of high molecular weight proteins during storage (especially for late storage), which caused a marked reduction in the solubility of rice components and resulted in increased hardness and reduced gumminess after storage. However, further investigations are needed to verify this observation.

**Cooking quality.** Except for the pH of the rice soup, there were no significant differences in the trend in cooking quality index during storage between different nitrogen application

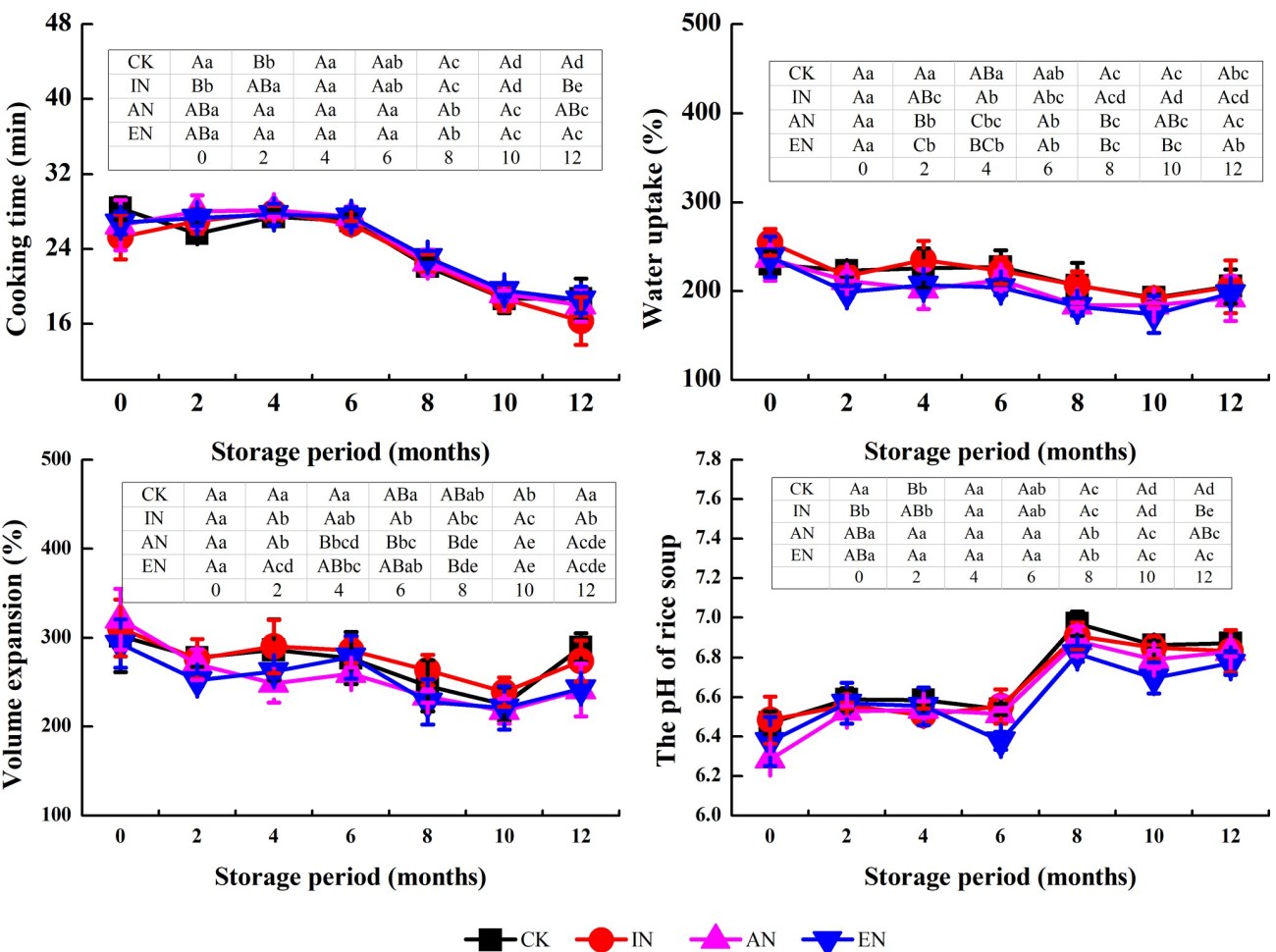

**Fig 5. Changes in the cooking quality of rice during storage, after the different nitrogen application rates.** CK, control group (0 kg N/ha); IN, insufficient nitrogen (160 kg N/ha); AN, adequate nitrogen (260 kg N/ha); EN, excessive nitrogen (420 kg N/ha). For each parameter, the mean ± standard deviation (n = 3) followed by different lowercase letters in the same rows differ significantly as a function of storage time (Duncan's test; $p < 0.05$). Different uppercase letters in the column denote significant differences as a function of different nitrogen application rates ($p < 0.05$).

rates (Fig 5), which indicated that nitrogen fertilizer application during cultivation exerted a less remarkable effect on the cooking quality of rice during storage.

The pH of rice soup between the four nitrogen treatments had little difference from 0 to 4 months of storage but had significant differences from 4 to 12 months of storage. Moreover, EN treatment exacerbated the reduction in the pH of rice soup during storage for 4–6 months. The changes in the pH of rice soup for CK, IN, AN, and EN groups were -0.7%, +0.7%, -0.3%, and -3%, respectively. Further, the EN treatment maintained a lower pH of rice soup even after storage of rice for 6 months, compared with the other treatments.

The main sources of acidic substances in rice during storage are the free fatty acids produced during fat oxidation [1]. EN may have increased the activity of enzymes related to lipid oxidation, such as lipoxygenase, and increased lipid oxidation to produce more fatty acids, resulting in a greater reduction in the pH of rice soup compared with the CK group. Although the change in pH of the rice soup was not enough to cause sensory differences, it may be possible that the changes in fatty acids, flavor substances, and other metabolites might be due to the change in pH, and this requires further investigation.

**Pasting characteristics of rice flour.** Pasting characteristics of rice flour are considered to be closely related to the ECQ of rice [34] and are also one of the sensitive indicators that reflect rice aging during storage [2]. The rate of decline in the viscosity index usually represents the aging of rice.

Small differences were observed between the four nitrogen application rates in terms of the changes in the setback, breakdown, peak time, and pasting temperature during storage (Fig 6). Further, the viscosity indices (peak viscosity, final viscosity, and trough viscosity) of rice in the AN and EN treatments were lower than that in CK during the one-year storage period, which indicated that AN and EN application were not conducive to maintaining good viscosity of rice. This is consistent with a previous study [35]. Nevertheless, the decline of trough viscosity started in rice after 4 months of storage for the CK and IN groups and after 6 months for the AN and EN treatments. For final viscosity, the trend was similar to that observed with the trough viscosity. It implied that AN and EN application delayed the aging of rice during storage to some extent.

Although we only discuss the change in pasting characteristics over one year, considering that the storage time of rice in actual production may be longer than this, we believe that if the storage is prolonged, this effect due to the delay may be more obvious. Therefore, this observation may be of considerable significance in delaying the change in pasting characteristics during rice aging in the future, and the underlying mechanisms require further study. Additionally, excessive nitrogen application is detrimental to the environment, such as GHG emissions [36], N leaching leading to pollution of waterways, and eutrophication [37]. However, we found that excessive nitrogen might be able to delay the aging of rice. This may provide a new basis for evaluation of the trade-off between the positive and negative effects in agriculture that occur by excessive nitrogen application.

## Correlation between rice quality index and changes in chemical composition

In all experimental groups, the effect of protein/amylose ratio on the ECQ of rice was greater than that of the amylose and protein contents alone (Table 1), complementing the previous observation reported upon analyzing amylose and protein contents and their correlation with the ECQ of rice [38,39]. We observed that the protein/amylose ratio had a greater impact on the eating quality of rice during storage, implying that this index must be considered when investigating the changes in rice quality during future storage operations. Furthermore, the protein/amylose ratio had a significantly negative correlation with breakdown and the peak and trough viscosity ($p < 0.05$; Table 1). Apparently, the change in peak and trough viscosity of rice, delayed by excessive nitrogen application (Fig 6), might be related to the change in protein/amylose ratio. In this study, this ratio was at a higher level in the EN treatment (S1 Fig), which implied that a high protein/amylose ratio may be beneficial in the stability of rice. Nonetheless, the observation warrants further investigation and validation in other rice varieties.

## Conclusions

To the best of our knowledge, this is the first report on the effect of nitrogen fertilizer levels on rice quality post-storage. Results showed that excessive nitrogen application significantly increased the hardness and reduced the gumminess during late storage. AN and EN applications delayed the change in the viscosity of rice paste. Additionally, during storage, the protein/amylose ratio showed a better correlation to the ECQ of rice than the protein or amylose contents alone. The results demonstrated that fertilizer application rate affected the quality of the agricultural product during storage by influencing the initial quality of the raw material.

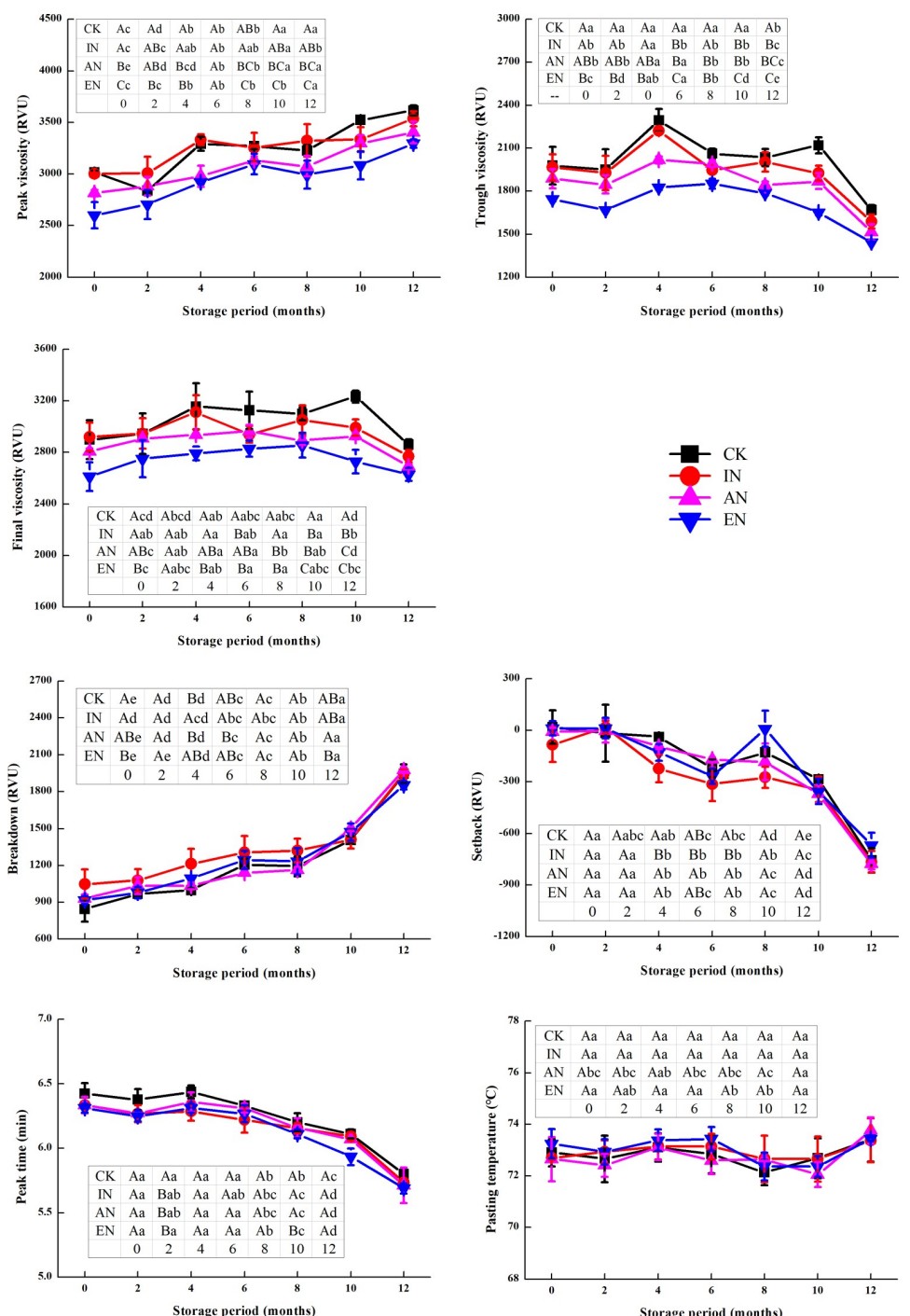

**Fig 6. The changes in pasting characteristics of rice during storage, after different nitrogen application rates.** CK, control group (0 kg N/ha); IN, insufficient nitrogen (160 kg N/ha); AN, adequate nitrogen (260 kg N/ha); EN, excessive nitrogen (420 kg N/ha). Data (mean ± standard deviation, n = 9) with different letters are significantly different (p < 0.05). For each parameter, different lowercase letters in the same rows differ significantly as a function of storage time. Different uppercase letters in the column denote significant differences as a function of different nitrogen application rates.

**Table 1. Correlation analysis between taste quality indices and chemical composition changes of rice during storage, following different nitrogen application rates.**

| | Protein content | | | | Amylose content | | | | Protein content/amylose content | | | |
|---|---|---|---|---|---|---|---|---|---|---|---|---|
| | N0 | N160 | N260 | N420 | N0 | N160 | N260 | N420 | N0 | N160 | N260 | N420 |
| Taste value | -.893** | | | | | | | | -.848* | -.848* | -.848* | -.848* |
| Hardness | | | | | | -.893** | | | | | | |
| Gumminess | | -.786* | | | | | .786* | | | | | |
| Springiness | | | | -.775* | | | .964** | | -.818* | -.818* | -.818* | -.818* |
| Cooking time | | | | -.757* | .811* | | | | -.798* | -.798* | -.798* | -.798* |
| Water uptake | | | | -.883** | .893** | | | | -.815* | -.815* | -.815* | -.815* |
| Volume expansion | | -.821* | | -.955** | | | | | | | | |
| Rice soup pH | | .786* | | .883** | | | | | | | | |
| Peak viscosity | | | | | | | | | | -.891** | -.820* | -.805* |
| Trough viscosity | | | | | | | | | | -.855* | -.904** | -.886** |
| Breakdown | | | | | | | | | | -.855* | -.907** | -.888** |
| Setback | | | | | | | | | | -.777* | | |
| Peak time | | | -.821* | -.818* | | | | | | -.794* | | -.776* |

*p < 0.05

**p < 0.01.

Hence, we should focus on the effects of pre-harvest field fertilization on the quality changes of agricultural products during storage to better ensure the high quality of agricultural products from the field to the plate.

## Supporting information

**S1 Fig. The changes in protein/amylose ratio of rice during storage, after different nitrogen application rates.** CK, control group (0 kg N/ha); IN, insufficient nitrogen (160 kg N/ha); AN, adequate nitrogen (260 kg N/ha); EN, excessive nitrogen (420 kg N/ha). Data
(mean ± standard deviation, n = 9) with different letters are significantly different (p < 0.05). For each parameter, different lowercase letters in the same rows differ significantly as a function of storage time. Different uppercase letters in the column denote significant differences as a function of different nitrogen application rates.
(TIF)

**S1 Table. Significance of variance estimates related to the interactions between storage time and nitrogen application rates on the quality traits of rice grain.**
(DOCX)

## Acknowledgments

The authors appreciate the contributions of Tian Zhang, Chen Yang, Yubo Liu, Wanning Zhao, Yu Fu, Tianyu Wang, Liyan Rong, and Baiyu Gu in the execution of the experiments.

## Author Contributions

**Conceptualization:** Zhaoxia Wu.

**Data curation:** Hanling Liang.

**Formal analysis:** Hanling Liang.

**Funding acquisition:** Zhaoxia Wu, Wentao Sun.

**Investigation:** Hanling Liang, Dongbing Tao, Qi Zhang, Shuang Zhang, Lifei Liu.

**Methodology:** Hanling Liang, Dongbing Tao, Qi Zhang, Shuang Zhang, Jiayi Wang.

**Project administration:** Hanling Liang, Jiayi Wang, Zhaoxia Wu, Wentao Sun.

**Resources:** Zhaoxia Wu, Wentao Sun.

**Supervision:** Zhaoxia Wu.

**Writing – original draft:** Hanling Liang.

**Writing – review & editing:** Hanling Liang, Zhaoxia Wu.

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
