## [Decision Letter · Decision Letter 0]

7 Apr 2021

PONE-D-21-06634

Nitrogen fertilizer application rate impacts eating and cooking quality of rice after storage

PLOS ONE

Dear Dr. Liang,

Thank you for submitting your manuscript to PLOS ONE. After careful consideration, we feel that it has merit but does not fully meet PLOS ONE’s publication criteria as it currently stands. Therefore, we invite you to submit a revised version of the manuscript that addresses the points raised during the review process.

Please carefully follow the journal guidelines as we hope we can minimize the number of times the document needs to be returned to you for further editing and proofreading.

please consider the following reviewers’ comments.

We look forward to receiving your revised manuscript.

Kind regards,

Walid Elfalleh, Ph.D

Academic Editor

PLOS ONE

Journal Requirements:

Reviewers' comments:

Reviewer's Responses to Questions

**Comments to the Author**

1. Is the manuscript technically sound, and do the data support the conclusions?

Reviewer #1: Yes

Reviewer #2: Partly

Reviewer #3: Yes

Reviewer #4: Partly

2. Has the statistical analysis been performed appropriately and rigorously? 

Reviewer #1: Yes

Reviewer #2: Yes

Reviewer #3: Yes

Reviewer #4: Yes

3. Have the authors made all data underlying the findings in their manuscript fully available?

Reviewer #1: Yes

Reviewer #2: Yes

Reviewer #3: Yes

Reviewer #4: Yes

4. Is the manuscript presented in an intelligible fashion and written in standard English?

Reviewer #1: Yes

Reviewer #2: Yes

Reviewer #3: Yes

Reviewer #4: Yes

5. Review Comments to the Author

Reviewer #1: The research carried out is correct although very simple. If it is on line with the editorial policy of PLOS ONE, it is ok for me.

The minor observations that I have made to this paper are included in the text.

Reviewer #2: I have not reviewed this manuscript previously and thought that the manuscript was very well written, with very few English language issues. Overall I think the concept and findings of how the addition of nitrogen fertiliser effects the quality of rice after storage, is good and worthy of publication. I have some minor queries in relation to the experimental set up and the need for some amendments to the text. Firstly sample size - where the separate fertiliser treatments grown in separate plots (4 fertiliser treatments times by X number of plots?), or is there more pseudo replication within the experiment one plot / area for each of the fertiliser treatments, with all the ECQ experiments tested on subsamples from the one plot? What area was each N treatment grown over in the field? Secondly, although I know you are concentrating on post-harvest, I feel a few lines need to be added to the discussion regarding the environmental consequences of your findings... Excessive nitrogen is bad for the environment - GHG emissions, N leaching leading to pollution of waterways, eutrophication etc BUT if excessive nitrogen delays quality change in rice, does that reduce the likelihood of food waste?

Minor comments

In your rebuttal to previous reviewers you use the phrase "post-harvest storage" I think you should use this phrase more within the manuscript (Abstract, keywords etc) as it conveys the essence of your study better.

line 81: it is standard to present organic matter as a percentage rather than g/kg, please amend

line 87: amend text to "except for the nitrogen application rates, standard practice for rice cultivation procedures were followed by local farmers"

line 114 and 116: should be author and then year only in brackets - e.g. in the study by Champagne et al. (1999)

line 193-194: Need to reword - besides, without storage the taste value of rice under EN treatment decreased doesn't make sense.

Reviewer #3: The manuscript presented a considerable improvement in the light of the reviewers’ comments; however, there are still some points to consider:

Line 109, 114, 116, 123 : « ….according to Champagne et al. (Champagne et al., 1996) »

-« …according to Champagne et al. (1996) »

Line 161 :« Nitrogen application did not impact the change of protein, fat, and moisture contents

in rice during storage »

- This is incorrect for protein content as was clearly reflected by the figure and the statistical analysis. Nitrogen application affects significantly this parameter, whereas, for fat and moisture parameters, there is, in general, no effect of N treatment but a significant effect of storage time. Please verify.

Line 168 : « …in the study by Gu et al., and was attributed to endogenous degradation (Gu et al., 2019). »

-« …in the study by Gu et al. (2019) and was attributed to endogenous degradation. »

Figures 3, 4, 5: « different lowercase letters in the same column differ significantly as a function of storage time as per Duncan’s test (p< 0.05). Different uppercase letters in the rows denote significant differences as a function of nitrogen application rates… »

-It is rather the reverse for column and rows (lowercase for rows and uppercase for column). Please verify.

-For Figs 3, 4, 5 and 6: it is recommended to conserve the same color for the same treatment.

Line 242 : « Except for the pH of the rice soup, there were no significant differences in the changes in cooking quality index during storage between different nitrogen application rates (Fig 5), »

- According to the statistical analysis, there is a significant effect for N application in the four parameters in fig 5. Please verify.

Line 247: “The changes in the pH of rice soup for CK, IN, AN, and EN groups were -0.7%, +0.7%, -0.3%, and -3%, respectively »

- If the comparison concerning the month 0 and the month 12, there is an increase with all the treatments. If it is between 0 and 6 months, verify again especially for AN treatment.

Line 249: “Further, the EN treatment maintained a low pH of rice soup even after storage of rice for 6 months. »

-If the interpretation concerned a specific time (at 6 months), the information was already noted in the line 246. If it is about the total storage time, there is a significant increase in pH from 6 months. Please verify.

-Line 309 : Fig. 6

Reviewer #4: Some general comments:

The language is not correct and in some part not understandable.

Line 53 to 59: add references.

Line 84: In Materials section, you mentioned that all treatments with insufficient (IN; 160 kg 85 N/ha), adequate (AN; 260 kg N/ha), and excessive (EN; 420 kg N/ha) nitrogen, based on the typical nitrogen fertilization of 260 kg N/ha used by local farmers. I think that it is insufficient to specify the group of treatment, you have to add a scientific reference.

Line 89: some details in the methods are missing and need to be clarified.

In the text, cite the reference number in square brackets as required by the journal.

Revise the list of references, the instructions on format and style of references are not at all followed.

6. PLOS authors have the option to publish the peer review history of their article (what does this mean?). If published, this will include your full peer review and any attached files.

Reviewer #1: No

Reviewer #2: No

Reviewer #3: **Yes: **Dr. Najet Gammoudi

Reviewer #4: **Yes: **Ahlem ZRIG

---

## [Author Response · Author response to Decision Letter 0]

17 Apr 2021

Dear Editor and reviewers, 

We wish to express our gratitude to the four reviewers for their professional opinions and suggestions. And we have made various modifications to the manuscript as much as we can, based on the opinions and suggestions of the four reviewers. The queries of the four reviewers have been answered as follows:

Journal Requirements:

Thank you for your valuable comments. We have revised the format of the manuscript according to the relevant documents.

Thank you for your suggestion. We have submitted the relevant data as supplementary materials. 

Reviewer #1: 

The research carried out is correct although very simple. If it is on line with the editorial policy of PLOS ONE, it is ok for me. The minor observations that I have made to this paper are included in the text.

We appreciate the comment put forth by the reviewer. We have responded to all questions in this article. Please see the text for details.

Reviewer #2: 

1. Firstly sample size - where the separate fertiliser treatments grown in separate plots (4 fertiliser treatments times by X number of plots?), or is there more pseudo replication within the experiment one plot/area for each of the fertiliser treatments, with all the ECQ experiments tested on subsamples from the one plot? What area was each N treatment grown over in the field? 

We thank the reviewer for their suggestion. We have added the relevant contents in the section of materials and methods: “The size of each subplot was 50 m2. The transplanting density was 30 cm × 18.2 cm with three seedlings placed in each hill. The experiment was conducted in three replicates” in lines 81-82.

2. Secondly, although I know you are concentrating on post-harvest, I feel a few lines need to be added to the discussion regarding the environmental consequences of your findings... Excessive nitrogen is bad for the environment - GHG emissions, N leaching leading to pollution of waterways, eutrophication etc BUT if excessive nitrogen delays quality change in rice, does that reduce the likelihood of food waste?

We appreciate the suggestion by the reviewer. I have added relevant contents at the end of the discussion: “Additionally, excessive nitrogen application is detrimental to the environment, such as GHG emissions [1], N leaching leading to pollution of waterways, and eutrophication [2]. However, we found that excessive nitrogen might be able to delay the aging of rice. This may provide a new evaluation basis of the trade-off between the positive and negative effects in agriculture that occur by excessive nitrogen application.”, in lines 269-274. 

Minor comments

3. In your rebuttal to previous reviewers you use the phrase "post-harvest storage" I think you should use this phrase more within the manuscript (Abstract, keywords etc) as it conveys the essence of your study better.

Considering the reviewer’s suggestion, we have revised the relevant content in this article.

4. line 81: it is standard to present organic matter as a percentage rather than g/kg, please amend 

Thank you for your comments. We have revised the relevant contents as follows: “The physical and chemical properties of the 0–20-cm soil were as follows: pH 8.2, organic matter 2.26%, total nitrogen 0.14%, alkali nitrogen 10.52%, available phosphorus 0.002%, available potassium 0.016%, and bulk density 1.39 g/cm3.”, in lines 73-75.

5. line 87: amend text to "except for the nitrogen application rates, standard practice for rice cultivation procedures were followed by local farmers"

We are grateful to the reviewer for the helpful suggestions. We have modified it to " Except for the nitrogen application rates, local farmers applied implemented practice for rice cultivation procedures." in lines 79-81.

6. line 114 and 116: should be author and then year only in brackets - e.g. in the study by Champagne et al. (1999)

We apologize for our negligence on this issue. We have respected your opinion and modified the reference format of the manuscript according to the PLoS One reference format.

7. line 193-194: Need to reword - besides, without storage the taste value of rice under EN treatment decreased doesn't make sense.

Thank you for your opinion. We would like to clarify that compared with the change in taste value of rice before and after one year of storage, the changes were greater between different nitrogen application rates. Therefore, I have revised the original text as follows: “Rice exposed to different nitrogen application rates showed similar variation trends in taste value during storage (Fig 4). The taste value of rice was significantly reduced with a higher nitrogen application rate and slightly decreased after storage, which is consistent with previous studies [3, 4]. Moreover, without storage, the taste value of rice under EN treatment significantly decreased from 63 to 48 compared to the CK group. For the CK group, the taste value of rice stored for 12 months slightly decreased from 63 to 60 compared to that observed without storage (Fig 4). Notably, nitrogen application had a greater impact on the taste value of rice than storage under the conditions of this study. Accordingly, controlling the rate of nitrogen application for cultivating rice is particularly important for maintaining the high eating quality of rice during storage.”, in lines 179-181. 

Reviewer #3: 

The manuscript presented a considerable improvement in the light of the reviewers’ comments; however, there are still some points to consider:

1. Line 109, 114, 116, 123 : « ….according to Champagne et al. (Champagne et al., 1996) »-« …according to Champagne et al. (1996) »

Thank you for your suggestion. We have modified the reference format according to the format prescribed by PLoS One.

2. Line 161: « Nitrogen application did not impact the change of protein, fat, and moisture contents in rice during storage »- This is incorrect for protein content as was clearly reflected by the figure and the statistical analysis. Nitrogen application affects significantly this parameter, whereas, for fat and moisture parameters, there is, in general, no effect of N treatment but a significant effect of storage time. Please verify.

We apologize for this issue. It is true that nitrogen application significantly affects this parameter, whereas, for fat and moisture parameters, there is, in general, no effect of N treatment except for a significant effect of storage time. However, we focused on the differences in the change trends of those parameters during storage between different nitrogen application rates. We found that the degree change of those parameters during storage was not significantly related to nitrogen application rates. Therefore, we performed a change to the original text, hoping to express our observation more clearly. “Nitrogen application did not impact the change trend of protein, fat, and moisture contents in rice during storage (Fig 3).”, in lines 153-154.

3. Line 168: « …in the study by Gu et al., and was attributed to endogenous degradation (Gu et al., 2019). »-« …in the study by Gu et al. (2019) and was attributed to endogenous degradation. »

Thank you for your suggestion. The original text has been revised as follows: “The decline in amylose content of rice during storage was also observed by Gu et al. (2019) [5] and was attributed to endogenous degradation.”, in lines 159-160.

4. Figures 3, 4, 5: « different lowercase letters in the same column differ significantly as a function of storage time as per Duncan’s test (p< 0.05). Different uppercase letters in the rows denote significant differences as a function of nitrogen application rates… »-It is rather the reverse for column and rows (lowercase for rows and uppercase for column). Please verify.

Thank you for your careful reading and scrutiny of our manuscript: “CK, control group (0 kg N/ha); IN, insufficient nitrogen (160 kg N/ha); AN, adequate nitrogen (260 kg N/hm2); EN, excessive nitrogen (420 kg N/ha). Data (mean ± standard deviation, n = 9) with different letters are significantly different (p < 0.05). For each parameter in the inset table, different lowercase letters in the same rows differ significantly as a function of storage time (Duncan’s test; p < 0.05). Different uppercase letters in the column denote significant differences as a function of nitrogen application rates (p < 0.05).”, in lines 170-175, 218-223, 245-250, 277-282.

5. For Figs 3, 4, 5 and 6: it is recommended to conserve the same color for the same treatment.

We thank the reviewer for highlighting this. The color of all figures has been changed.

6. Line 242: « Except for the pH of the rice soup, there were no significant differences in the changes in cooking quality index during storage between different nitrogen application rates (Fig 5), »- According to the statistical analysis, there is a significant effect for N application in the four parameters in fig 5. Please verify.

We thank the reviewer for their queries and suggestions. With the prolongation of the storage period, these four indices do have significant changes. However, we focused on the effect of nitrogen application rate on the change range and trend of those indices before and after storage. It can be observed that different nitrogen application rates only have a significant effect on the change trend and range of pH during storage. We changed the original text to: “Except for the pH of the rice soup, there were no significant differences in the trend in cooking quality index during storage between different nitrogen application rates (Fig 5), which implied that nitrogen fertilizer application during cultivation had little effect on the cooking quality of rice during storage.”, in lines 226-229. 

7. Line 247: “The changes in the pH of rice soup for CK, IN, AN, and EN groups were -0.7%, +0.7%, -0.3%, and -3%, respectively »- If the comparison concerning the month 0 and the month 12, there is an increase with all the treatments. If it is between 0 and 6 months, verify again especially for AN treatment.

Thank you for your opinion, which is similar to the opinion expressed in Question 6. As we focused on the changes of pH during storage under different nitrogen treatments, we showed that the differences mainly occurred at 4-6 months. However, we clarified this by the following modifications: “The pH of rice soup between the four nitrogen treatments had little difference from 0 to 4 months of storage but had significant differences from 4 to 12 months of storage. Moreover, during the entire storage period, EN treatment exacerbated the reduction in the pH of rice soup during storage for 4–6 months. The changes in the pH of rice soup for CK, IN, AN, and EN groups were -0.7%, +0.7%, -0.3%, and -3%, respectively.”, at lines 230-235. 

8. Line 249: “Further, the EN treatment maintained a low pH of rice soup even after storage of rice for 6 months. »-If the interpretation concerned a specific time (at 6 months), the information was already noted in the line 246. If it is about the total storage time, there is a significant increase in pH from 6 months. Please verify.

We express our gratitude to the reviewer for the advice. I have revised the original text: “Further, the EN treatment maintained a lower pH of rice soup even after storage of rice for 6 months compared to other treatments.”, at lines 234-235.

9. -Line 309: Fig. 6

Thank you for your careful reading of my manuscript. We have modified it according to your suggestion, in line 294.

Reviewer #4: 

Some general comments: The language is not correct and in some part not understandable.

1. Line 53 to 59: add references.

Thank you for your valuable comments. Here we have changed to “Low temperature and humidity [6, 7], as well as vacuum or nano packaging [8], have been proven to be beneficial in maintaining rice quality during storage. Such methods have been applied in milled rice that has a higher commodity price[9, 10]; however, such methods have not been applied in rice paddy owing to the trade-off between the high cost of those storage methods and the relatively low commodity value of rice paddy. Thus, rice paddy is usually stored under natural conditions[11].” at lines 48-53.

2. Line 84: In Materials section, you mentioned that all treatments with insufficient (IN; 160 kg 85 N/ha), adequate (AN; 260 kg N/ha), and excessive (EN; 420 kg N/ha) nitrogen, based on the typical nitrogen fertilization of 260 kg N/ha used by local farmers. I think that it is insufficient to specify the group of treatment, you have to add a scientific reference.

We appreciate the reviewer’s concern. In this study, the treatment of three nitrogen rates is selected according to our previous research results; hence, we have added the following contents in the materials and methods section: “The following treatment groups were established based on our previous results [12] due to their significant effect on the ECQ of rice: a control group without nitrogen treatment (CK; 0 kg N/ha), and three treatment groups with insufficient (IN; 160 kg N/ha), adequate (AN; 260 kg N/ha), and excessive nitrogen (EN; 420 kg N/ha) treatment. The typical nitrogen fertilization of 260 kg N/ha was usually used by local farmers.”, at lines 75-79.

3. Line 89: some details in the methods are missing and need to be clarified. In the text, cite the reference number in square brackets as required by the journal.

We thank the reviewer for their comment. To explain the experimental methods in more detail, the following contents have been added: “All harvested grains were air-dried for a month to reduce moisture content to approximately 14%; each rice paddy (500 g) was packed in a nylon net bag, placed in a carton, and stored under laboratory conditions for 12 months.”, at lines 86-88.

4. Revise the list of references, the instructions on format and style of references are not at all followed.

Thank you for your suggestion. we have modified the reference format of the full text according to the reference format of PLoS One.

We thank you again for your time and hope that you will find the article to be suitably modified.

Sincerely, 

Hanling liang

References 

1. Alvaro-Fuentes J, Luis Arrue J, Cantero-Martinez C, Isla R, Plaza-Bonilla D, Quilez D. Fertilization Scenarios in Sprinkler-Irrigated Corn under Mediterranean Conditions: Effects on Greenhouse Gas Emissions. Soil Science Society of America Journal. 2016;80(3):662-71. doi: 10.2136/sssaj2015.04.0156. PubMed PMID: WOS:000378848900012.

2. Hamonts K, Balaine N, Moltchanova E, Beare M, Thomas S, Wakelin SA, et al. Influence of soil bulk density and matric potential on microbial dynamics, inorganic N transformations, N2O and N-2 fluxes following urea deposition. Soil Biol Biochem. 2013;65:1-11. doi: 10.1016/j.soilbio.2013.05.006. PubMed PMID: WOS:000323686800001.

3. Zhu DW, Zhang HC, Guo BW, Xu K, Dai QG, Wei HY, et al. Effects of nitrogen level on yield and quality of japonica soft super rice. J Integr Agric. 2017;16(5):1018-27. doi: 10.1016/s2095-3119(16)61577-0.

4. Liu HJ, Watanabe K, Tojo S, Sugiyama T, Makino E. A study on the effect of storage conditions upon rice quality (Part 1) Change in quality of milled rice during storage. Journal of the Japanese Society of Agricultural Machinery. 2002.

5. Gu F, Gong B, Gilbert RG, Yu W, Li E, Li C. Relations between changes in starch molecular fine structure and in thermal properties during rice grain storage. Food Chem. 2019;295:484-92. doi: 10.1016/j.foodchem.2019.05.168.

6. Genkawa T, Uchino T, Inoue A, Tanaka F, Hamanaka D. Development of a low-moisture-content storage system for brown rice: Storability at decreased moisture contents. Biosystems Engineering. 2008;99(4):515-22. doi: 10.1016/j.biosystemseng.2007.12.011.

7. Park CE, Kim YS, Park KJ, Kim BK. Changes in physicochemical characteristics of rice during storage at different temperatures. Journal of Stored Products Research. 2012;48(48):25-9. doi: 10.1016/j.jspr.2011.08.005.

8. Wang F, Hu Q, Mariga AM, Cao C, Yang W. Effect of nano packaging on preservation quality of Nanjing 9108 rice variety at high temperature and humidity. Food Chem. 2018;239:23-31. doi: 10.1016/j.foodchem.2017.06.082. PubMed PMID: WOS:000408740200004.

9. Ahmad U, Alfaro L, Yeboah-Awudzi M, Kyereh E, Dzandu B, Bonilla F, et al. Influence of milling intensity and storage temperature on the quality of Catahoula rice ( Oryza sativa L.). LWT-Food Sci Technol. 2017;75:386-92.

10. Kim OW. The Quality of Milled Rice with Reference to Whiteness and Packing Conditions during Storage. 2007;14(1):18-23. PubMed PMID: KJD:ART001043654.

11. Nath S. Indoor Storage of Paddy-Rice in the Lowlands of Papua New Guinea. Ama-Agricultural Mechanization in Asia Africa and Latin America. 2009;40(1):46-9. PubMed PMID: WOS:000266287800009.

12. Liang HL, Gao SY, Ma JX, Zhang T, Wang TY, Zhang S, et al. Effect of Nitrogen Application Rates on the Nitrogen Utilization, Yield and Quality of Rice. Food and Nutrition Sciences. 2020;12:15. Epub 27. doi: 10.4236/fns.2021.121002

---

## [Decision Letter · Decision Letter 1]

18 May 2021

PONE-D-21-06634R1

Nitrogen fertilizer application rate impacts eating and cooking quality of rice after storage

PLOS ONE

Dear Dr. Liang,

Thank you for submitting your manuscript to PLOS ONE. After careful consideration, we feel that it has merit but does not fully meet PLOS ONE’s publication criteria as it currently stands. Therefore, we invite you to submit a revised version of the manuscript that addresses the points raised during the review process.

The paper seems improved and can be considered by the journal, however the reviewers raised some minor comments. Please revise accordingly.

We look forward to receiving your revised manuscript.

Kind regards,

Walid Elfalleh, Ph.D

Academic Editor

PLOS ONE

Journal Requirements:

Reviewers' comments:

Reviewer's Responses to Questions

**Comments to the Author**

1. If the authors have adequately addressed your comments raised in a previous round of review and you feel that this manuscript is now acceptable for publication, you may indicate that here to bypass the “Comments to the Author” section, enter your conflict of interest statement in the “Confidential to Editor” section, and submit your "Accept" recommendation.

Reviewer #1: All comments have been addressed

Reviewer #3: (No Response)

Reviewer #4: All comments have been addressed

2. Is the manuscript technically sound, and do the data support the conclusions?

Reviewer #1: Yes

Reviewer #3: Yes

Reviewer #4: Yes

3. Has the statistical analysis been performed appropriately and rigorously? 

Reviewer #1: Yes

Reviewer #3: Yes

Reviewer #4: Yes

4. Have the authors made all data underlying the findings in their manuscript fully available?

Reviewer #1: Yes

Reviewer #3: Yes

Reviewer #4: Yes

5. Is the manuscript presented in an intelligible fashion and written in standard English?

Reviewer #1: Yes

Reviewer #3: Yes

Reviewer #4: Yes

6. Review Comments to the Author

Reviewer #1: The paper can now be accepted for publication since the authors have modified their previous text according to my corrections.

Reviewer #3: “The decline in amylose content of rice during storage was also observed by Gu et al. (2019) [5] and was attributed to endogenous degradation”

-Delete the year and parentheses

-The reference number does not match that in the revised manuscript. Verify.

-Insert the revised reference list in the manuscript and verify again all the correspondence author-number.

Reviewer #4: Line 42: rewrite this sentence, so long

Line 50-53: rewrite this sentence, not clear

Line 153-167: add more percentages to describe the results

Line 394: revise this reference.

Line 397: some reference, delete it. 22. National food safety standards. Determination of fat in food. Beijing, China:National Health and Family Planning Commission; 2016.

23. National food safety standards. Determination of moisture in food. Beijing, China:

National Health and Family Planning Commission; 2016.

7. PLOS authors have the option to publish the peer review history of their article (what does this mean?). If published, this will include your full peer review and any attached files.

Reviewer #1: No

Reviewer #3: **Yes: **Dr. Najet Gammoudi

Reviewer #4: **Yes: **ZRIG Ahlem

---

## [Author Response · Author response to Decision Letter 1]

25 May 2021

May 25, 2021

PLOS ONE

Dear Editor and Reviewers, 

We wish to re-submit the manuscript titled “Nitrogen fertilizer application rate impacts eating and cooking quality of rice after storage.” The manuscript ID is PONE-D-21-06634R1.

We wish to express our gratitude to the three reviewers for their professional opinions and suggestions. We have made various modifications to the manuscript to the best of our ability based on the opinions and suggestions of the three reviewers. The queries of the three reviewers have been answered below.

We thank you again for your time and hope that you will find the article to be suitably modified.

Sincerely,

Hanling Liang

College of Food Science, Shenyang Agricultural University, Shenyang 110866, People’s Republic of China

 

Journal Requirements:

Response: Thank you for your careful reading and scrutiny of our manuscript. We have reviewed the reference list and ensured that it is complete and correct. We have added three references as mentioned below in the response to the 2nd comment from Reviewer 4 and removed a reference as mentioned in response to the 4th comment from Reviewer 4.

Comments to the author

Reviewer’s responses to questions

Reviewer #1:

The paper can now be accepted for publication since the authors have modified their previous text according to my corrections.

Response: We appreciate the comment put forth by the reviewer.

Reviewer #3:

1. “The decline in amylose content of rice during storage was also observed by Gu et al. (2019) [5] and was attributed to endogenous degradation”

-Delete the year and parentheses

Response: Thank you for your comments. We have deleted the relevant content.

2. The reference number does not match that in the revised manuscript. Verify.

-Insert the revised reference list in the manuscript and verify again all the correspondence author-number.

Response: We are grateful to the reviewer for the helpful suggestions. We have verified all the reference numbers within the manuscript and checked the correspondence between authors and numbers.

Reviewer #4: 

1. Line 42: rewrite this sentence, so long

Response: Thank you for your comments. We have revised the relevant content as follows (lines 40–42):

However, several studies have reported that the storage process may result in the deterioration of the eating quality of the rice, such as increased hardness and reduced viscosity [1].

2. Line 50-53: rewrite this sentence, not clear

Response: We thank the reviewer for their suggestion. We have rewritten the relevant content as follows and added references 11–13 for clarification (lines 49–53):

Such methods have been applied in milled rice that has a higher commodity price [11, 12]; however, such methods have not been widely applied in the storage of paddy rice, owing to the trade-off between the high cost of those storage methods and the relatively low commodity value of paddy rice. Thus, paddy rice is usually stored under natural conditions [13]. 

3. Line 153-167: add more percentages to describe the results

Response: Thank you for your comments. We have revised the relevant content as follows (lines 155–163):

Nitrogen application did not impact the change trend of protein, fat, and moisture contents in rice during storage (Fig 3). The change in rice content before and after storage under the four nitrogen application treatments had a range of 8%–12% for protein, 5%–11% for fat, and 29%–30% for moisture. Amylose content in the CK and IN samples did not vary significantly during storage; however, EN treatment significantly exacerbated the rate of amylose decline during the period of storage, spanning from 4 to 10 months. During the storage period of 4–10 months, the amylose content decreased by 1% in the CK samples and 4% with EN treatment, which is approximately four times that of CK. This finding demonstrated that nitrogen application in the field changed the starch metabolism of rice during storage.

4. Line 394: revise this reference.

Response: Thank you for your careful reading and scrutiny of our manuscript. This reference describes our previous research results that we published in Food and Nutrition Sciences. Because the journal is not included in SCI, it may not be found on Web of Science and other channels, but the article can be retrieved through Google Scholar or by the DOI. Here, in order to avoid misunderstanding, I deleted this reference and made relevant modifications in the materials and methods as follows (lines 75–82):

The following treatment groups were established based on our previous research, in which the effect of nitrogen application rates (0, 160, 210, 260, 315, and 420 kg N/ha) on rice quality was investigated, and due to significant effects on the ECQ of rice: a control group without nitrogen treatment (CK; 0 kg N/ha) and three treatment groups with insufficient (IN; 160 kg N/ha), adequate (AN; 260 kg N/ha), and excessive nitrogen (EN; 420 kg N/ha) treatment. The typical nitrogen fertilization of 260 kg N/ha was usually used by local farmers. Except for the nitrogen application rates, standard practices for rice cultivation were followed by local farmers.

5. Line 397: some reference, delete it. 22. National food safety standards. Determination of fat in food. Beijing, China:National Health and Family Planning Commission; 2016.

23. National food safety standards. Determination of moisture in food. Beijing, China:

National Health and Family Planning Commission; 2016.

Response: Thank you for your comments. We have deleted the relevant content.

---

## [Editor Report · Decision Letter 2]

31 May 2021

Nitrogen fertilizer application rate impacts eating and cooking quality of rice after storage

PONE-D-21-06634R2

Dear Dr. Wu,

We’re pleased to inform you that your manuscript has been judged scientifically suitable for publication and will be formally accepted for publication once it meets all outstanding technical requirements.

Kind regards,

Walid Elfalleh, Ph.D

Academic Editor

PLOS ONE
---

## [Editor Report · Acceptance letter]

10 Jun 2021

PONE-D-21-06634R2 

Nitrogen fertilizer application rate impacts eating and cooking quality of rice after storage 

Dear Dr. Wu:

I'm pleased to inform you that your manuscript has been deemed suitable for publication in PLOS ONE. Congratulations! Your manuscript is now with our production department. 

Kind regards, 

on behalf of

Professor Walid Elfalleh 

Academic Editor

PLOS ONE